**communications** engineering

**Comment**

# Scorecard for synthetic medical data evaluation

Ghada Zamzmi, Adarsh Subbaswamy, Elena Sizikova, Edward Margerrison, Jana G. Delfino & Aldo Badano

Although the interest in synthetic medical data (SMD) for developing and testing artificial intelligence (AI) methods is growing, the absence of a comprehensive framework to evaluate the quality and applicability of SMD hinders its wider adoption. Here, we outline an evaluation framework designed to meet the unique requirements of medical applications. We also introduce SMD scorecard, a comprehensive report accompanying artificially generated datasets. This scorecard provides a quantitative assessment of SMD across seven criteria (7 Cs), complemented by a descriptive section that contains all relevant information about the dataset. The SMD scorecard provides a practical framework for evaluating and reporting the quality of synthetic data, which can benefit SMD developers and users.

A key challenge for the safe and effective development of medical artificial intelligence (AI) devices is the limited availability of patient data[1], as data sharing is often restricted due to well-founded privacy concerns. Further, data collection is time-consuming, costly, and sometimes unfeasible for rare diseases and underrepresented populations (e.g., data on Caucasians with sickle cell disease or Black with skin cancer). Synthetic medical data (SMD), partially or fully generated artificial data using computational techniques to mimic the properties and relationships seen in patient data[2], holds promise to address these challenges.

SMD has gained attention due to recent advances in generative deep learning techniques. Methods such as Generative Adversarial Networks (GANs) and Denoising Diffusion Probabilistic Models have the ability to approximate the distributions of medical data and create synthetic distributions that align with patient data. These methods can generate large-scale medical data to supplement the limited patient data available for medical AI development. However, the usefulness of SMD hinges on its quality. If SMD is not carefully curated, it could result in poor outcomes for downstream tasks: a classic case of "garbage in, garbage out".

Therefore, a systematic approach is needed for evaluating and reporting the quality of SMD to enable its proper use in medical applications. Current methods[3] for assessing synthetic data are inadequate because they do not holistically evaluate the generated data based on medically relevant criteria.

The evaluation strategies commonly used are based on metrics developed initially for computer vision tasks, focusing mainly on statistical fidelity[3]. This focus on fidelity alone fails to consider the unique and complicated aspects of medical data. For example, Fig. 1 displays synthetically generated medical images that perform well on known metrics such as Fréchet Inception Distance (FID) and statistical similarity measures. Nevertheless, these images fail to meet constraints and exhibit artifacts such as broken ligaments, misplaced pacemakers, or multiple nipples. Although these artifacts may pass undetected by existing evaluation metrics, they risk misleading AI models trained or tested on such data, potentially resulting in incorrect predictions.

The figure also highlights current issues with evaluating synthetic textual data generated by large language models (LLMs). In the medical query task, the LLM provided incorrect answers. As current evaluation metrics[4] (e.g., BLEU and ROUGE) are designed to assess the similarity between generated text and ground truth responses based on n-gram overlap, these metrics can successfully detect explicit mismatches, such as incorrect answers, by measuring how closely the generated text matches the reference text. However, these same metrics do not account for hallucinated or fabricated information[4], as demonstrated in the summarization task. In this task, the LLM introduces a hallucinated symptom (leg swelling) that is not present in the original Electronic Health Record (EHR) note. This error exposes a fundamental limitation of BLEU and ROUGE as they focus on overlap and recall of n-grams without evaluating the factual accuracy or contextual relevance of the generated text. Even if the hallucinated symptom is medically irrelevant or non-sensical in the given context, these metrics do not identify such discrepancies because they do not analyze the content for clinical constraints. This limitation poses risks in clinical applications. For example, a clinician relying on the summary might assume the presence of a symptom explicitly denied by the patient, which can potentially lead to misdiagnosis or inappropriate treatment. These examples underscore the need for comprehensive evaluation frameworks that go beyond fidelity and statistical similarity to incorporate dimensions such as clinical constraints (e.g., clinical validity and anatomical accuracy).

In this paper, we present a framework (Fig. 2) for evaluating and reporting SMD based on medically relevant criteria and introduce SMD scorecard. This scorecard would accompany an artificially generated medical dataset and provide quantitative assessments of SMD quality across multiple criteria while documenting key information such as dataset description, intended use, limitations, recommendations, and usage disclaimers. This framework can advance evaluation, transparency, and the responsible use of synthetic datasets by guiding data developers, data providers, and data users in adopting standardized practices for SMD evaluation and reporting.

## The 7 Cs of SMD Evaluation

> When a measure becomes a target, it ceases to be a good measure.
> – Goodhart's Law

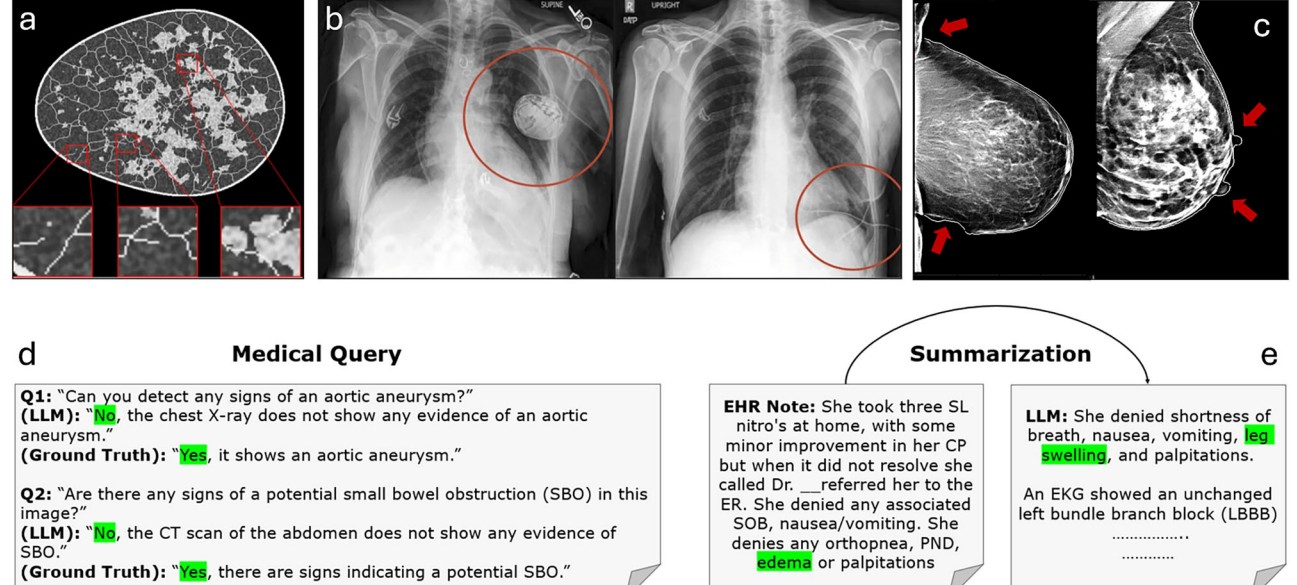

**Fig. 1 | Evaluation challenges in synthetic medical images and textual data. a** A synthetic image produced by a generative AI model (adopted from Ref. 17 with high visual fidelity but containing structural inconsistencies such as broken ligaments, **b**, **c** Synthetic chest X-rays ([18] featuring misplaced medical devices (pacemakers and tubes) located outside of anatomically plausible regions, and (**c**) digital mammograms that score highly on fidelity and statistical metrics but display clinically implausible artifacts including abnormal breast shape and the erroneous presence of multiple nipple-like structures indicated by the red arrows. (**d**) shows textual outputs from LLMs in a medical query task where the LLM provides incorrect responses to critical clinical questions[19] which can often be detected through direct comparison with ground truth. However, in the summarization task (**e**), the LLM generates a symptom (leg swelling) not present in the original EHR note, which is harder to detect using similarity- or overlap-based metrics.

In the context of SMD, this law underscores the risks of focusing exclusively on a single measure, such as alignment between synthetic and patient data. While such alignment is important, over-optimizing for it can result in neglecting other criteria. To address this, we propose evaluating SMD using a set of comprehensive criteria and measures. This section presents the "7 Cs": **C**ongruence, **C**overage, **C**onstraint, **C**ompleteness, **C**ompliance, **C**omprehension, and **C**onsistency. We note that while this set of criteria is comprehensive, it is not exhaustive; additional criteria may be included as necessary.

Ensuring the quality of synthetic data is important for its successful use in both training and testing AI models. In the context of training, synthetic data can help enlarge datasets and address class imbalances during model development. However, poor-quality data can undermine these benefits by amplifying biases, embedding irrelevant features, and confusing models, ultimately impairing their ability to generalize to real-world scenarios. For testing, synthetic datasets should reflect real-world diversity and heterogeneity to avoid overconfidence in model performance. Without this, models evaluated on synthetic data may fail in real-world deployments and compromise patient safety. To mitigate these risks, proactive evaluation of synthetic data quality, whether for training or testing, is needed. It enables researchers to identify and address potential issues early in the process. To this end, we propose a framework for the systematic and proactive evaluation of SMD to guide its use and maintain its quality.

While we recognize the value of qualitative approaches for assessing SMD, quantitative approaches may drive consistency. We also note that the aim of such evaluation is not to identify the 'best' synthetic dataset. Rather, the aim is to establish a transparent and comprehensive framework for assessing dataset quality across multiple clinically relevant criteria. Figure 2 provides a visual illustration of the challenges, and Table 1 presents a template of the SMD scorecard.

**Congruence**. Congruence assesses the degree to which the distribution of synthetic data aligns with the distribution of patient data. High Congruence indicates that synthetic samples closely match real ones in statistical properties and perceptual quality. In the literature, this concept is related to realism, which emphasizes perceptual quality, and fidelity, which focuses on the statistical alignment of generated and real data distributions[3,5,6].

Although Congruence alone is not sufficient to assess SMD, ensuring alignment between synthetic and patient data is a necessary first step before evaluating other aspects. Congruence can be evaluated using several metrics tailored to different data types[3,5]. For image data, metrics such as Cosine Similarity and FID assess alignment in the feature space and compare the distributions of synthetic and real images, respectively. For textual data, Cosine Similarity measures alignment between text embeddings, while metrics like the BLEU score evaluate similarity based on n-gram overlap between synthetic and reference texts[4]. These metrics are summarized under the Congruence dimension in Table 2.

**Coverage**. Coverage evaluates the extent to which SMD captures the variability, range, and novelty inherent in patient data while aligning with true distributions. This evaluates whether SMD represents the variability and breadth of patterns, features, and modes present in patient data. Novelty evaluates whether the dataset contains unique samples that introduce new and valid variations while remaining aligned with real data plausibility.

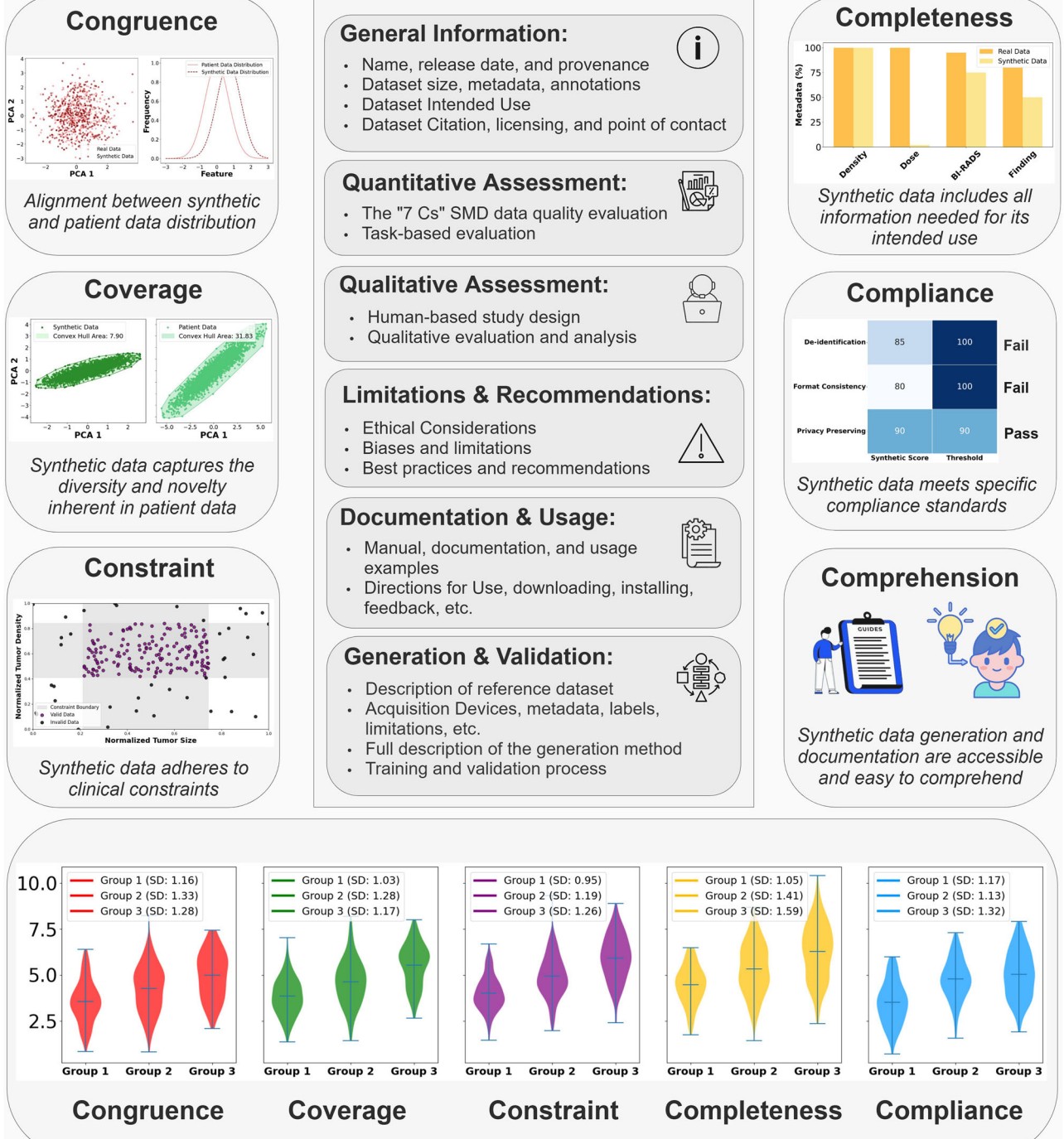

**Fig. 2 | Scorecard for evaluating and reporting synthetic medical data.** Congruence measures the alignment between the distributions of synthetic and real data. Coverage highlights the variability in synthetic data, demonstrated using convex hull volume, where synthetic data shows a smaller spread compared to patient data. Constraint evaluates adherence to clinical context (e.g., tumor size and density) by identifying valid versus invalid data points. Completeness assesses the presence of necessary information for the intended use. Compliance ensures adherence to local and global standards such as de-identification, privacy preservation, and file format consistency. Comprehension evaluates the accessibility and clarity of SMD generation processes and documentation. The bottom panel illustrates the consistency across subgroups (e.g., demographic or disease-specific groups). Essential descriptive information about the synthetic data is presented in the middle panel.

**Table 1 | Synthetic Medical Data (SMD) scorecard Template**

| 1. Synthetic Data General Information | |
|---|---|
| Name | [Dataset name] |
| Release Date | [Release date] |
| Version History | [Version numbers and updates] |
| Dataset Size | [Size, including number of samples, cases, and patients] |
| Dataset Modality | [Data modality, e.g., imaging (CT, MRI, X-ray), textual] |
| Dataset Provenance | [Origin and generation method of the dataset] |
| Dataset Intended Use | [Primary and secondary applications and use cases] |
| Dataset Labels | [Details of labels including types (e.g., binary, categorical, segmentation)] |
| Attribution and Licensing | [Citation information and license details] |
| Point of Contact | [Contact details for inquiries] |
| **2. Data Quality Evaluation (7 Cs Quantitative Results)** | |
| Congruence | [Evaluation of alignment between synthetic and real data] |
| Coverage | [Evaluation of feature and population representativeness] |
| Constraint | [Evaluation of adherence to known constraints (e.g., clinical or technical)] |
| Completeness | [Evaluation of missing or incomplete information in the synthetic data] |
| Compliance | [Evaluation of adherence to ethical, clinical, or healthcare standards] |
| Comprehension | [Evaluation of interpretability or transparency, if applicable] |
| Consistency | [Evaluation of data quality across batches, subgroups, or over time] |
| **3. Task-based Evaluation (Quantitative Results)** | |
| Task Performance | [Evaluation of the synthetic dataset's effectiveness for a specific task] |
| Task-Specific Metrics | [Metrics such as sensitivity, or mean IoU along with acceptance thresholds] |
| **4. Human-based Evaluation (Qualitative Results)** | |
| Human Study Design | [Summary of the reader study design, including evaluator expertise] |
| Reader Study Results | [Results of human evaluation (e.g., realism, and clinical relevance) |
| Observations & Failure Cases | [Insights from qualitative evaluation, highlighting strengths, weaknesses, or failure cases] |
| **5. Ethical, Legal, and Practical Considerations** | |
| Privacy & Anonymization | [Details on anonymization, privacy, and compliance with standards] |
| Biases | [Known biases, their potential impacts on dataset use, and applied or suggested mitigation strategies] |
| Limitations | [Other technical and clinical limitations of the dataset] |
| Recommendations | [Recommendations and best practices] |
| **6. Synthetic Dataset Usage** | |
| Repository Access | [Direct link to the dataset repository, DOI, and any access requirements] |
| Preprocessing Requirements | [Details of data format and preprocessing steps required for use] |
| User Documentation | [Links to README files, user guides, tutorials, or other supporting materials for dataset usage] |
| Intended Audience | [Description of target users (e.g., researchers, clinicians, developers)] |
| **7. Synthetic Dataset Training & Validation Process** | |
| Generation Method | [Description of the synthetic data generation pipeline] |
| Training & Validation Process | [Details of the training and validation process] |
| **8. Reference Dataset General Information** | |
| Purpose | [Description of the reference dataset's purpose and primary use cases] |
| Origin & Source | [Origin of the reference dataset, sites, and dates from which data was collected] |
| Dataset Size | [Details on the size including number of samples, cases, or patients] |
| Clinical Population | [Characteristics of the clinical population, such as demographics, disease prevalence, and inclusion/exclusion criteria] |
| Acquisition Devices | [Details of the devices and parameters used for data acquisition] |
| Reference Standard | [Details of the reference standard, including annotation methods and qualifications of annotators] |
| Ground Truth Labels | [Details of ground truth annotations available] |

**Table 1 (continued) | Synthetic Medical Data (SMD) scorecard Template**

| 1. Synthetic Data General Information | |
|---|---|
| Metadata | [Available metadata such as age, gender, breast density, etc.] |
| Preprocessing | [Preprocessing steps applied to the dataset] |
| Known Limitations | [Known limitations of the reference dataset such as biases or technical constraints] |

Section 2 (Data Quality Evaluation) presents the quantitative results based on the 7 Cs introduced in this work.

This criterion is applicable to various data types[3,5]. For image data, metrics such as Convex Hull Volume and Clustering-Based metrics can quantify Coverage by examining the spread and distribution of data points within the feature space. For textual data, Recall can assess Coverage by measuring the proportion of distinct terms or patterns in synthetic notes compared to real notes. For numerical data, metrics like Variance and Entropy evaluate the range and dispersion of synthetic values, ensuring they span the real-world distribution while introducing novel but medically valid combinations. Other metrics can be found in Table 2.

**Constraint.** We define Constraint as the extent to which SMD respects known constraints, which may include anatomical, biological, temporal, geometric, clinical, or user-defined constraints.

Constraint violations can occur in both imaging and non-imaging synthetic data. For instance, in the summarization task (Figure 1), a hallucinated symptom is introduced that is absent from the original EHR note. In digital mammography (Figure 1), maintaining anatomical accuracy requires adherence to constraints such as correct breast shape and the appropriate number of nipples. Additionally, enforcing constraints such as valid size and density in synthetic tumor images or ensuring valid lab value ranges in generated medical reports is important for making SMD suitable for training diagnostic tools.

This criterion adherence can be quantified using various metrics. For example, the Constraint Violation Rate[7] (see Table 2) measures the frequency at which generated data deviates from predefined clinical constraints or fails to align with relevant symptoms for specific medical conditions. Other metrics, such as Nearest Invalid Datapoint and Distance to Constraint Boundary[7], evaluate the proximity of synthetic data points to invalid data points or clinically established boundaries, ensuring that generated data remains within acceptable ranges In some cases, these constraint boundaries can be directly established from patient data to reflect real-world distributions. Other metrics, such as those evaluating structural and anatomical plausibility (e.g., maintaining realistic tissue density or geometric accuracy), are detailed in[8,9].

**Completeness.** Completeness is defined as the extent to which generated data contains all necessary details relevant to the task. In medical imaging, Completeness can refer to whether essential information, such as metadata, annotations, and clinical details, is included when required for the application. For textual data, Completeness ensures that the generated output provides complete information and findings without omitting key elements essential for the clinical task. In tabular data, Completeness involves including all relevant features and records needed for the clinical task; missing data or incomplete records would lead to lower Completeness.

Given a reference standard for comparison, several metrics can quantify Completeness. For example, the Proportion of Required Fields (Table 2) measures the fraction of essential fields present in the synthetic data relative to the reference dataset. Missing Data Percentage quantifies the proportion of missing values compared to the total number of expected values. Additionally, scaling-based metrics provide a measure of Completeness by rating how well the generated data captures essential details on a defined scale (e.g., 1 to 10). These metrics quantify the extent to which SMD preserves the necessary details found in patient data, which help in identifying missing information that could compromise the usability of the SMD.

**Compliance.** We define Compliance as the reporting of adherence of SMDs to established format guidelines (e.g., DICOM for medical imaging), privacy standards, and relevant local and international requirements. Compliance is particularly important in data-driven methods that might suffer from leakage of protected patient information. For example, GANs and LLMs are designed to generate SMD that closely approximates real-world datasets. However, when these models are trained on extensive datasets without adequate safeguards, they may inadvertently reproduce outputs containing sensitive patient information[10]. A significant privacy concern arises when the generated data is overly similar to specific real-world entries, potentially replicating identifiable details from the training data and leading to privacy violations.

Metrics to measure privacy compliance include Differential Privacy metrics, which evaluate the level of privacy protection in the synthetic data, and Re-identification Risk metrics, which assess the likelihood of identifying individuals in the synthetic data. Also, audit trails and compliance reports can be used to continuously monitor and document the adherence of SMD generation to relevant standards. See Table 2 for examples of Compliance metrics, and refer to[10] for further details on privacy-preserving methods in synthetic data generation.

We recognize that privacy and other requirements vary across regions and institutions, and the Compliance criterion is broad enough to accommodate a variety of contexts. Despite regional differences, Compliance can be consistently evaluated using the metrics in Table 2 with region-specific thresholds. For instance, some regions may require stricter thresholds (e.g., lower $\varepsilon$ or higher $k$) to minimize re-identification risks, while other regions allow more flexible thresholds. Regardless of the region, the same Compliance metrics can be applied, with thresholds adjusted to meet specific requirements.

**Comprehension.** Comprehension refers to how easily users can understand the process of generating SMD. This criterion evaluates the transparency and clarity of the data generation process as well as the quality of the accompanying documentation. One metric to evaluate the quality of documentation is the Documentation Clarity Score (Table 2). This metric quantifies how clearly the generation process is documented by using scale-based methods.

Knowledge-based approaches for SMD generation typically achieve higher Comprehension because parameters such as tumor sizes, noise ranges, or dose levels are explicitly defined and well-documented. These methods often include detailed annotations and examples that directly link the synthetic data to underlying clinical rules or constraints, making them

**Table 2 | Examples of metrics to evaluate SMD quality across seven dimensions**

| Metric Example | Space | Binary? | Direction | Image Metric? |
|---|---|---|---|---|
| **Congruence** | | | | |
| Cosine Similarity | Embedding | Binary | Maximize | No |
| Earth Mover's Distance | Embedding | Binary | Minimize | No |
| Jensen-Shannon Divergence | Embedding | Binary | Minimize | No |
| Peak Signal-to-Noise Ratio | Image | Binary | Maximize | Yes |
| Structural Similarity Index | Image | Binary | Maximize | Yes |
| Fréchet Inception Distance | Embedding | Binary | Minimize | Yes |
| Distance to Centroid | Embedding | Binary | Minimize | No |
| Precision | Embedding | Binary | Maximize | No |
| **Coverage** | | | | |
| Inception Score | Image | Unary | Maximize | Yes |
| Recall | Embedding | Binary | Maximize | No |
| Coverage | Embedding | Binary | Maximize | No |
| Convex Hull Volume | Embedding | Unary | Maximize | No |
| Determinantal Point Processes Score | Embedding | Unary | Maximize | No |
| Vendi Score | Embedding | Unary | Maximize | No |
| Variance | Embedding | Unary | Maximize | No |
| Entropy | Embedding | Unary | Maximize | No |
| Clustering-Based Metrics | Embedding | Unary | Maximize | No |
| **Constraint** | | | | |
| Nearest Invalid Datapoint | Embedding | Binary | Minimize | No |
| Distance to Constraint Boundary | Embedding | Binary | Minimize | No |
| Constraint Violation Rate | Embedding | Binary | Minimize | No |
| **Completeness** | | | | |
| Proportion of Required Fields | Metadata | Binary | Maximize | No |
| Missing Data Percentage | Metadata | Binary | Minimize | No |
| **Compliance** | | | | |
| Differential Privacy Score | Data Attribute | Unary | Minimize | No |
| K-Anonymity Level | Data Attribute | Unary | Maximize | No |
| L-Diversity Score | Data Attribute | Unary | Maximize | No |
| T-Closeness Level | Data Attribute | Unary | Maximize | No |
| **Comprehension** | | | | |
| Documentation Clarity Score | Documentation | Unary | Maximize | No |

**Table 2 (continued) | Examples of metrics to evaluate SMD quality across seven dimensions**

| Metric Example | Space | Binary? | Direction | Image Metric? |
|---|---|---|---|---|
| **Consistency** | | | | |
| Variance | Quality Metrics | Unary | Minimize | No |
| Maximum-Minimum Difference | Quality Metrics | Unary | Minimize | No |
| Analysis of Variance | Quality Metrics | Unary | Stat. Sig. | No |

The table includes the following columns: **Space**, indicating the domain where the metric operates (e.g., embedding, metadata fields); **Binary?**, specifying whether the metric can be applied to synthetic data alone (unary) or requires comparison with patient data (binary); **Direction**, describing the optimization goal (e.g., maximize, minimize, or statistical significance); and **Image-Metric?**, which indicates if the metric is restricted to image, or applicable to other types, such as clinical notes or numerical data. Note that **Direction** of Coverage metrics can be maximized within a specific range to ensure SMD contains sufficient variability while remaining aligned with patient data. For further details about these metrics and how they are measured, refer to[3,7,16].

inherently more transparent and easier to understand and document. In contrast, generative AI methods, which rely on implicit processes, often score lower on Comprehension due to challenges in interpreting hidden parameters and the complex, opaque nature of their algorithms.

**Consistency**. Consistency refers to the stability of SMD quality metrics across different subgroups or over time. This criterion evaluates whether SMD consistently meets quality criteria (e.g., Congruence, Coverage, Constraint, Completeness) across a range of patient demographics, disease classes, or other subgroups. Ensuring uniform quality across these groups prevents disparities that could negatively impact downstream applications. Consistency also applies over time, ensuring that SMD quality remains stable as real-world data landscapes evolve. This is particularly important for applications such as longitudinal studies and disease progression modeling, where temporal stability in synthetic data is needed for reliable predictions.

To assess Consistency, metrics such as Variance, Maximum-Minimum Difference, and Analysis of Variance (ANOVA) (Table 2) can be employed. While Variance and Maximum-Minimum Difference quantify average and extreme variations across subgroups or time points, ANOVA is useful for statistically assessing whether the variability observed between different subgroups or conditions is significant.

**SMD Scorecard**

Inspired by the Model Scorecard[11] and Healthsheet[12], we propose the Synthetic Data Scorecard. While the proposed scorecard shares similarities with the Model Scorecard and Healthsheet in promoting documentation and standardization, it is uniquely tailored for synthetic datasets, emphasizing quality evaluation through the "7 Cs" framework alongside comprehensive reporting of dataset characteristics and limitations. This scorecard may serve as both a quality assurance tool and a resource for stakeholders including developers and users to better understand the characteristics, quality, and limitations of SMD.

As shown in Table 1, the proposed scorecard includes the following sections.

1. **General Information**: This section provides basic details about the dataset, such as modality, size, labels, licensing, and point of contact for inquiries.

2. **Data Quality Evaluation (Quantitative)**: The core of the proposed scorecard, referred to as the Scorecard, evaluates the dataset against the 7 Cs criteria. This section of the scorecard presents detailed results on data quality and provides quantitative results for each dimension.

3. **Task-based Evaluation (Quantitative)**: This section provides quantification of the dataset's utility in a specific task using task-specific metrics.

4. **Human-based Evaluation (Qualitative)**: This section provides results from qualitative evaluations. It includes description of study design, results of the reader study, and a summary of identified weaknesses or failure cases.

5. **Ethical Considerations, Limitations, and Recommendations**: This section highlights ethical concerns including known biases and limitations. Recommendations for best practices are provided.

6. **Usage**: This section offers guidance on utilizing the dataset. It includes links to repositories or DOIs, details on preprocessing, documentation (e.g., README or user guides).

7. **Synthetic Dataset Training and Validation Process**: This section documents the processes used for generating and validating the synthetic dataset.

8. **Reference Dataset Information**: This section outlines key details of the patient dataset used for comparison, including its size, population characteristics, and known limitations. Such information is particularly important for generative methods, as it forms the foundation for assessing the quality and relevance of the synthetic dataset.

The scorecard, combined with the "7 Cs" framework, can benefit various stakeholders. Data developers can use the proposed scorecard to provide comprehensive documentation and evaluation for their datasets. Researchers can use the scorecard to decide whether the SMD is suitable for specific applications or tasks based on its characteristics, including alignment with patient data characteristics, adherence to application-specific constraints, consistency across subgroups, compliance with relevant standards, and completeness to ensure no task-related information is missing. To facilitate the creation of the scorecard, we created a template that simplifies the process of completing the descriptive sections. Additionally, we will release a Python library to compute the quantitative aspects of the scorecard, focusing on the metrics associated with the 7 Cs framework.

## Discussion

The proposed SMD scorecard, along with its "7 Cs" quantitative framework, offers a multimodal approach for evaluating the quality of SMD. Beyond establishing distribution alignment between synthetic and patient data (Congruence), the framework evaluates SMD based on other criteria. Coverage measures the variability of synthetic data to ensure it reflects real-world heterogeneity and trends. Constraint assesses adherence to known constraints (e.g., clinical, technical, geometrical). These three criteria address alignment, representation, and adherence to constraints. The framework also includes Completeness to assess whether the dataset includes all necessary data elements required for its intended application. Compliance describes adherence to privacy requirements and standards. Consistency reports data quality uniformity across subgroups or over time. Lastly, Comprehension delineates the presence and characteristics of dataset documentation.

### Applicability to Other Modalities

This framework can be extended to different contexts and data types beyond imaging. For example, for clinical notes, Congruence can assess alignment using semantic features (e.g., token embeddings) and similarity metrics, while for tabular data, it compares statistical distributions of key attributes. Coverage can evaluate variants in clinical notes via token variability, and in tabular data through statistical spread tests.

Similarly, Constraint can be applied to other data types. For example, in medical imaging, this criterion ensures anatomical plausibility, such as verifying that breast mass sizes and shapes align with clinically plausible ranges using features like shape descriptors or spatial consistency checks. For clinical notes and tabular data, constraints can be derived from diagnostic guidelines or known physiological ranges. For instance, an important constraint in clinical notes is ensuring consistency in symptom-condition relationships, which can be achieved through graph modeling of extracted tokens to verify that symptom descriptions align with the documented condition. In numerical data, constraints can assess whether values fall within known physiological ranges; for example, hemoglobin levels should be within the range of normal or condition-specific thresholds (e.g., ranges for anemia versus normal levels). By defining valid constraint boundaries-whether anatomical, clinical, or physiological-synthetic data can be rigorously validated.

Other criteria, such as Completeness and Consistency, can also be generalized. Consistency is inherently adaptable, as it focuses on assessing the uniformity of quality criteria (e.g., Coverage, Constraints) across subgroups or over time. Completeness ensures the inclusion of all essential elements across data types: for imaging data, it can verify the presence of annotations and metadata; for textual data, it can ensure generated outputs include all critical information; and for numerical data, it validates the presence of all necessary variables and records. Finally, the Compliance criterion can emphasize core principles (e.g., data security and privacy) while adapting thresholds for compliance with global or regional requirements. For example, privacy and data anonymization methods may use consistent metrics globally, but thresholds or specific requirements can vary by region. This approach ensures the metrics remain applicable while accommodating regional requirements effectively.

### Feature Representation for Quantitative Assessment

The quantitative assessment of certain criteria using specific metrics (see Table 2) requires transforming the data into a feature embedding space. For images, common feature representations include radiomic features, deep features, topological features, and shape-based properties. These features capture various aspects of the image, such as texture, intensity distributions, shape, and geometric patterns, which are essential for meaningful comparisons and evaluations. For textual data, token embeddings are often used, which leverages natural language processing techniques to encode semantic and syntactic information into vector representations. For numerical data, statistical summaries (e.g., mean, variance, clustering features) can serve as key features. Since high-dimensional feature spaces may contain redundant or irrelevant information, dimensionality reduction techniques (e.g., principal component analysis) or feature summarization methods are often applied to condense the extracted features into compact embeddings. Such a careful representation of features is important for an accurate quantitative evaluation.

These constructed feature representations serve as the foundation for both global and local assessments of synthetic data quality. Global assessments ensure that the synthetic data maintains high overall quality, capturing aspects such as global distribution alignment and adherence to constraints. Local assessments, on the other hand, focus on specific regions of interest (e.g., localized lesion areas). For example, high global Congruence indicates overall alignment between synthetic and patient data distributions (e.g., images exhibit similar texture, contrast, and statistical properties).

High local Congruence, meanwhile, ensures that the characteristics of lesion areas in synthetic data closely align with those in patient data, capturing critical features that directly impact diagnostic accuracy. This distinction between global and local assessments helps identify and address potential blind spots in synthetic data quality. Specifically, while global assessments may reveal broad distribution-level alignment, local evaluations can uncover issues such as subtle inconsistencies in lesion texture or shape. Together, this hierarchical approach balances broad trends and nuanced details, providing a robust framework for evaluating the quality of synthetic data and ensuring its reliability for both training and testing AI models.

**Aggregating Scores for Decision Thresholds**. To obtain a simplified yet comprehensive evaluation for each criterion, individual metrics can be aggregated into a single score. Score aggregation is a widely used approach across various domains, including credit scoring, health indices, and risk assessments, and can be effectively applied here. By consolidating multiple metrics into a single representative value, score aggregation provides a concise summary of overall performance, making it easier to interpret results, compare datasets, and draw meaningful conclusions. Different aggregation methods can be employed depending on the evaluation context and the importance of individual metrics. For example, within Coverage, metrics can be combined using techniques such as normalized or weighted averages, geometric means, and other statistical or machine learning-based approaches (e.g., fuzzy aggregation)[13]. The choice of aggregation method may depend on the relative importance of each metric and the context of the evaluation. Weighted averages, for example, allow greater emphasis on metrics deemed critical for the specific clinical application.

Once the aggregated score is calculated, appropriate thresholds can be set. These thresholds may be informed by clinical knowledge or derived from a reference dataset. For example, if a reference patient dataset demonstrates a Coverage range between 80% and 85%, the following thresholds could classify Coverage as: good if the score ≥80%, moderate if 70% ≤ score <80%, and low if the score <70%. Note that thresholds will vary depending on the clinical application or task.

To ensure transparency and a more comprehensive analysis, we recommend reporting both the aggregated score and individual metrics for each criterion. Although the aggregated score simplifies interpretation, individual metrics provide deeper insights into specific areas of model performance, helping to identify strengths and weaknesses. This dual reporting approach supports better decision making, facilitates troubleshooting, and enhances the reliability of synthetic data evaluations.

**Importance of Evaluating SMD for Training and Testing**. Assessing the quality of synthetic medical data (SMD) is important for its effective use in both training and testing AI models. While synthetic data can address data scarcity, poor-quality synthetic data can distort feature distributions and mislead the learning process. This can cause models to learn irrelevant patterns, amplify existing biases, and fail to generalize effectively to real-world clinical scenarios.

The risks associated with synthetic data depend on its application. Flawed or unrepresentative synthetic training data can weaken a model's ability to generalize, leading to performance degradation when applied to real patient data. Similarly, synthetic test data that lacks realism or variability may result in misleading evaluations, reducing its reliability for assessing model performance.

The impact of poor-quality synthetic data was exemplified in the case of IBM's Watson for Oncology, where the system failed due to its reliance on synthetic data that did not adequately capture the complexities, heterogeneity, and variability of real patient data. Similarly, a recent study revealed that LLMs degrade when trained on recursively generated data, further emphasizing the need for rigorous scrutiny of synthetic data quality in both training and testing[14]. Another study highlighted that improperly curated synthetic data can compromise utility and negatively impact downstream classification performance[15].

To mitigate these risks, proactive evaluation of SMD quality is essential. Implementing systematic quality control mechanisms can help identify and address potential limitations in synthetic datasets early in the development process, ensuring that AI models trained on such data are safe and effective.

**Final Remarks**. With the growing reliance on synthetic data in medical and scientific research, the need for systematic approaches to evaluate its quality has become increasingly important. Synthetic data is widely used to augment datasets, address privacy concerns, and enhance model generalizability, but its effectiveness depends on its fidelity, representativeness, and adherence to known data constraints. Without standardized evaluation methodologies, the risk of using low-quality synthetic data in medical applications remains a concern. The proposed framework, along with the scorecard, provides a structured approach for assessing whether synthetic data meets essential quality requirements and aligns with its intended use. By offering a practical and quantitative approach, this framework enables innovators and practitioners to make informed decisions about the synthetic datasets.

## Disclaimers

The mention of commercial products, their sources, or their use in connection with material reported herein is not to be construed as either an actual or implied endorsement of such products by the Department of Health and Human Services. This is a contribution of the U.S. Food and Drug Administration and is not subject to copyright.

## Data availability

This work does not analyze or provide any dataset, as it does not rely on computational techniques but rather focuses on proposing a framework.

**Ghada Zamzmi, Adarsh Subbaswamy, Elena Sizikova, Edward Margerrison, Jana G. Delfino & Aldo Badano** [ORCID] [✉]
Office of Science and Engineering Laboratories, Center for Devices and Radiological Health, U.S. Food and Drug Administration Silver Spring, Maryland, USA. [✉]e-mail: aldo.badano@fda.hhs.gov

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

## Author contributions

G.Z. and A.B. conceived the idea. G.Z., A.S., E.S., J.G.D., E.M. and A.B. contributed to the framework formulation. All authors contributed to the final preparation and proofreading of the manuscript.

## Competing interests

The authors declare no competing interests.
