## [Transparent Peer Review file · Communications Engineering]

The Scorecard for Synthetic Medical Data Evaluation

Corresponding Author: Dr Aldo Badano

Version 0:

Reviewer comments:

Reviewer #1

(Remarks to the Author)

This paper introduces the use of scorecards to assess synthetic medical data. It uses 7 'C's to produce this scorecard that cover the key concepts required in assessing a dataset.

This is an important topic and the scorecard seems like a useful approach to providing simple guidance on dataset quality. The 7 'C's appear to cover many of the key issues in synthetic data assessment.

One issue I would like to see discussed is that of other types of data beyond images - which seems to be the sole focus of this paper. Medical data comes in many forms and these scorecards should naturally extend to medical notes and numerical clinical data. There is one sentence on LLMs in the discussion but I feel that more details in this section would make this a more useful paper in terms of covering the diversity of medical data. For example, a discussion on linking the specific issues with medical notes / clinical data to the scorecard approach.

Another issue is the universality of this approach. For example, regulations may depend on the use case, or the country that data is being used. As a result, the scorecard will change depending on these. This is not a major problem for the approach, but rather this should be explored and discussed within the discussion section.

Reviewer #2

(Remarks to the Author)

Scorecards for Synthetic Medical Data Evaluation and Reporting

The papers proposes scorecards, a systematic framework for generating synthetic medical data (SMD) which could be used for training and evaluating healthcare AI algorithms.

Given the increased use (and perhaps hype) about SMD it is crucial to evaluate its quality and I support the authors' effort towards creating this qualitative evaluation framework.

A couple of thoughts/suggestions that could be helpful to discuss:

- There is an important distinction between using SMD for training vs testing algorithms. In training, even noisy/not highly realistic data could still be helpful for the algorithm to be more robust. But using SMD for evaluation is ill-advised, as since the data is generated from the training distribution, in a sense the model would be overfitting to the training set.
- Perhaps it would be good to give some specific examples of for example "garbage in garbage" and exemplify by specific research studies where this went wrong. In fact while the scorecard intuitively makes sense to me, "healthcare AI algorithms" can include many things. Or is this limited to medical imaging, as Fig 1 might suggest? It would be helpful to see an example of a filled in model card for an existing dataset/algorithm, like in the Model Cards paper, or here <https://arxiv.org/abs/1803.09010>, or even specifically for health: <https://dl.acm.org/doi/fullHtml/10.1145/3531146.3533239>
- This is example is for Correctness, but I think I have similar concerns for all the principles. I agree that medical data is not small computer vision and that many metrics may not be appropriate. But the Correctness principle then talks about other metrics for matching distributions, it is unclear from the reading why these would be superior. Since the data is high-dimensional, I imagine it might be intractable to compare the distribution of images in their original resolution, and the data

samples are not limited. So it is estimations of the distributions that will be compared – which means that it would be potentially possible for the distributions to be matching.

- Correctness: I'm not sure I understand the relevance of BLEU score (LLMs), but perhaps that is because I do not understand the scope of data being covered as said in an earlier point
- For all the principles: the framework is qualitative but all the principles are exemplified by quantitative metrics, what is a concrete recommendation for how to turn multiple possible metrics into for example “low coverage” assessment?
- Who should be doing these evaluations? Given the level of for example correctness, coverage and documentation of some existing public, non-synthetic datasets, it seems that researchers are not incentivized to do this.

In conclusion I support this motivation and topic, but overall I feel like the details of the metrics (why these would work better, how to combine them), and of the practical adoption of the framework, are not sufficiently described.

Version 1:

Reviewer comments:

Reviewer #1

(Remarks to the Author)

I am happy with the additions to the text to address my concerns.

Reviewer #2

(Remarks to the Author)

Thank you for your response and incorporating my earlier comments. I still think the training/testing distinction could have been emphasized earlier and I would be interested to read more in-depth discussions on the points you added (perhaps with subsections in the Discussion). But I believe the work has added value to the community, so I'm happy to recommend publication.

Reply to the Reviewers

Re: Manuscript ID COMMS-24-0115-T

“Scorecards for Synthetic Medical Data Evaluation and Reporting”

Nature Communications Engineering

We thank the editor for handling our manuscript and both the editor and reviewers for their constructive comments, which helped us enhance the quality and clarity of our manuscript. We have addressed all the comments in the revised manuscript and used blue text to highlight the changes. Below, we provide a point-by-point response to each comment.

Reviewer 1: Overall Summary

This paper introduces the use of scorecards to assess synthetic medical data. It uses 7 'C's to produce this scorecard that cover the key concepts required in assessing a dataset.

This is an important topic and the scorecard seems like a useful approach to providing simple guidance on dataset quality. The 7 'C's appear to cover many of the key issues in synthetic data assessment.

Reviewer #1, comment #1

One issue I would like to see discussed is that of other types of data beyond images - which seems to be the sole focus of this paper. Medical data comes in many forms and these scorecards should naturally extend to medical notes and numerical clinical data. There is one sentence on LLMs in the discussion but I feel that more details in this section would make this a more useful paper in terms of covering the diversity of medical data. For example, a discussion on linking the specific issues with medical notes / clinical data to the scorecard approach.

Our response #1.1

We thank the reviewer for acknowledging the value of our work. We agree that the scorecard approach extends beyond medical images to other types of medical data, such as clinical notes and numerical clinical data. To address this, we have revised the manuscript to include examples illustrating how the scorecard can be applied to other forms of medical data. Additionally, we expanded the discussion to explain how each scorecard criterion is applicable to different data types.

In response to the reviewer's comment, we have made the following changes (highlighted in blue):

- Added an example of medical notes alongside medical images in Figure 1.
- Expanded the description of all criteria to make them general and applicable to other data types.
- Included three new paragraphs in the discussion (second, third, and fourth paragraphs) elaborating on how the same criteria can be applied to other forms of medical data.

We hope this revision clarifies that the scorecard approach is flexible to cover other types of medical data.

Reviewer #1, comment #2

Another issue is the universality of this approach. For example, regulations may depend on the use case, or the country that data is being used. As a result, the scorecard will change depending on these. This is not a major problem for the approach, but rather this should be explored and discussed within the discussion section.

Our response #1.2

We thank the reviewer for raising the important point of how regulations may vary depending on the use case or country. We agree that regulatory requirements can differ. To address this, we have revised Section 2 (Compliance) and Section 4 (fourth paragraph) to emphasize that the scorecard is designed to be adaptable. Just as different models may have unique performance requirements based on their specific applications, the scorecard criteria (such as Correctness, Coverage, and Compliance) can be applied across synthetic datasets but with tailored requirements based on the region or intended use case. Specifically, the Compliance criterion can be adjusted to meet local regulatory requirements. For example, Compliance in the U.S. may emphasize HIPAA standards while in Europe, the focus would shift to GDPR.

The Compliance criterion can be measured using the same metrics across regions and institutions while allowing flexibility in thresholds to meet specific requirements. For example, in privacy-preserving metrics like Differential Privacy (ϵ) and K-Anonymity, in contexts that require stricter standards, we might require lower ϵ values (e.g., $\epsilon = 0.5$) or higher k levels (e.g., $k \geq 10$) to minimize re-identification risks. In contrast, in other contexts with more flexible standards, we might permit higher ϵ values (e.g., $\epsilon = 2$) or lower k levels (e.g., $k = 5$) to balance utility and privacy. Similarly, standardization metrics, such as a DICOM Conformance Score, can universally evaluate adherence to medical imaging standards but emphasize different attributes depending on institutional priorities.

Reviewer 2, overall comment

The paper proposes scorecards, a systematic framework for generating synthetic medical data (SMD) which could be used for training and evaluating healthcare AI algorithms.

Given the increased use (and perhaps hype) about SMD it is crucial to evaluate its quality and I support the authors' effort towards creating this qualitative evaluation framework.

A couple of thoughts/suggestions that could be helpful to discuss:

Reviewer #2, comment #1

There is an important distinction between using SMD for training vs testing algorithms. In training, even noisy/not highly realistic data could still be helpful for the algorithm to be more robust. But using SMD for evaluation is ill-advised, as since the data is generated from the training distribution, in a sense the model would be overfitting to the training set.

Perhaps it would be good to give some specific examples of for example “garbage in garbage” and exemplify by specific research studies where this went wrong.

Our response #2.1

We appreciate the reviewer’s pointing out this important distinction between using SMD for training versus testing algorithms. We agree that the impact of synthetic data quality is more critical when used for testing due to overfitting. However, we believe it is still important to assess the quality of synthetic data used for training. We presented our reason below and we revised the manuscript (7th paragraph in the Discussion) to discuss the importance of ensuring the quality for training and testing.

Importance of assessing training data: While the quality of SMD is more critical when used for testing, it is also important to assess the quality of synthetic data used for training. Poor-quality synthetic data can mislead models by causing them to learn from irrelevant patterns or features, resulting in overfitting and poor generalization to real-world data. This issue was exemplified in the failure of IBM’s Watson for Oncology. This high-profile collapse was attributed to the reliance on synthetic data that failed to capture the complexities and variabilities of patient data. Similarly, a recent study (Shumailov et al.) revealed that LLMs tend to degrade when trained on recursively generated data, further emphasizing the need to scrutinize the quality of synthetic data used in training. Such proactive assessment of synthetic data prior to training is important for identifying potential limitations and addressing them early. Although synthetic data can introduce diversity, poor-quality data can negatively impact training by embedding unrealistic biases, confusing the model, or distracting it from learning relevant patterns. This highlights the importance of conducting pre-training quality checks.

- Shumailov, Iliia, et al. “AI models collapse when trained on recursively generated data.” *Nature* 631.8022 (2024): 755-759.
- Faheem, Hadiya, and Sanjib Dutta. “Artificial Intelligence Failure at IBM’Watson for Oncology’.” *IUP Journal of Knowledge Management* 21.3 (2023): 47-75.

Garbage in, Garbage out: In response, we have included specific examples in the revised manuscript to demonstrate how poor data quality negatively impacts machine learning models. These are highlighted in blue in the revised manuscript.

A notable real-world example of “garbage in, garbage out” is IBM’s Watson for Oncology, which performed poorly when deployed in clinical settings. One of the major reasons that leads to its failure, as reported in the paper below, was the use of synthetic datasets that lacked quality and failed to represent the variability of real-world data. Beyond this example, several research studies (e.g., Shumailov et al., Jordon et al., Ghassemi et al.) have shown how the quality of synthetic data can degrade utility and reduce downstream classification performance. This real world example along with these studies emphasize the principle of “garbage in, garbage out,” illustrating how poor-quality data can harm model performance.

This underscores the importance of proactive approaches to ensure the quality of generated synthetic data before its use in training or testing. Our proposed framework addresses this need by providing a systematic way to evaluate synthetic data quality, enabling early detection of potential issues.

- Shumailov, Iliia, et al. "AI models collapse when trained on recursively generated data." *Nature* 631.8022 (2024): 755-759.
- Faheem, Hadiya, and Sanjib Dutta. "Artificial Intelligence Failure at IBM'Watson for Oncology'." *IUP Journal of Knowledge Management* 21.3 (2023): 47-75.
- Cheng, Victoria, et al. "Can you fake it until you make it? impacts of differentially private synthetic data on downstream classification fairness." *Proceedings of the 2021 ACM Conference on Fairness, Accountability, and Transparency*. 2021.
- Jordon, James, et al. "Synthetic Data—what, why and how?." arXiv preprint arXiv:2205.03257 (2022).

Reviewer #2, comment #2

In fact while the scorecard intuitively makes sense to me, "healthcare AI algorithms" can include many things. Or is this limited to medical imaging, as Fig 1 might suggest? It would be helpful to see an example of a filled in model card for an existing dataset/algorithm, like in the Model Cards paper, or here <https://arxiv.org/abs/1803.09010>, or even specifically for health: <https://dl.acm.org/doi/fullHtml/10.1145/>

Our response #2.2

We thank the reviewer for raising these important points.

Scope beyond medical imaging: We have revised the manuscript to clarify how each criterion can be adapted to other data types. Specifically, we expanded Figure 1, elaborated on each criterion in Section 2 to illustrate these extensions, and added three paragraphs in the Discussion section.

While Figure 1 focuses on an example from medical imaging, we intend the scorecard to be adaptable and not limited to this domain. It can be applied across a variety of healthcare AI algorithms, including large language models (LLMs) for clinical notes, or numerical clinical data, and more. The criteria—such as Correctness, Coverage, and Compliance—are broad enough to evaluate AI systems across diverse healthcare contexts. This mirrors the flexibility of the model card framework, which is designed to be general and adaptable, with task-specific nuances.

Comparison to Model Card: As discussed in Section 3 of the revised manuscript, the Synthetic Medical Data (SMD) Card distinguishes itself from Healthsheets and Model Card by focusing explicitly on reporting the characteristics and quality (7Cs framework) of synthetic datasets. While the main concept of documenting and standardizing information about datasets or models is shared, our SMD Card is uniquely tailored to synthetic data. It emphasizes evaluating the quality of synthetic medical data across technical, clinical, and regulatory dimensions, providing a systematic and multi-modal approach.

Unlike Healthsheets or Model Cards, which typically focus on broader datasets or model characteristics, the SMD Card integrates quantitative metrics and numerical results to assess the dataset's performance in areas such as Correctness, Coverage, Completeness, and Constraint. These metrics provide actionable conclusions, allowing users to determine the suitability of synthetic datasets for specific applications. Further, similar to how the main sections and criteria for Model Cards and Healthsheets remain consistent across applications while details like sensitivity or specificity acceptance thresholds vary, the SMD Card maintains consistent core criteria but adapts its details to align with the requirements of synthetic data in different tasks.

SMD Card example: As per the reviewer's suggestion, we have made the following revisions: 1) a template of the SMD scorecard is now provided in Table 1 of the revised manuscript; 2) an example of a completed scorecard, using the Sinkove synthetic digital mammography dataset, has been included in our webpage (https://github.com/alzanzmigaa/SMD_ScoreCard/tree/main/Sinkove_Example) to demonstrate its application; we did not provide the example in the manuscript as this is a perspective paper with limited word count; and 3) an example of our internal tool for generating the scorecard is shown on the project's webpage. Please note that this tool will be made publicly available.

Reviewer #2, comment #3

This is example is for Correctness, but I think I have similar concerns for all the principles. I agree that medical data is not small computer vision and that many metrics may not be appropriate. But the Correctness principle then talks about other metrics for matching distributions, it is unclear from the reading why these would be superior. Since the data is high-dimensional, I imagine it might be intractable to compare the distribution of images in their original resolution, and the data samples are not limited. So it is estimations of the distributions that will be compared – which means that it would be potentially possible for the distributions to be matching.

Our response #2.3

We thank the reviewer for their valuable feedback. In response, we clarified (Discussion section) that while Correctness (or Congruence) focuses on measuring alignment between synthetic and real data distributions to ensure a baseline level of similarity and alignment, other criteria, such as Coverage, extend beyond alignment to assess variability that reflects real-world diversity. Additionally, Constraint evaluates adherence to established clinical constraints specific to the medical task or modality. For example, Congruence ensures that synthetic data aligns with the statistical properties of real data, Coverage then evaluates the diversity and representativeness of the synthetic data using, and Constraint measures the adherence of synthetic data to known constraints, such as anatomical accuracy. For instance, Constraint could quantify whether tumors in synthetic images exhibit clinically accurate properties such as appropriate size, location, and shape, etc. Together, these criteria (along with the other four criteria) ensure a holistic assessment of synthetic data quality beyond distribution-matching between synthetic and patient data.

We also discussed the importance of extracting clinically relevant features instead of using the original resolution followed by using dimensionality reduction or feature summarization.

Detailed response: We agree that medical data differs from standard computer vision tasks, making direct comparisons of image distributions in their original resolution impractical. Metrics should instead focus on carefully extracted relevant features rather than raw image resolutions or generic features, ensuring evaluations are computationally feasible and clinically meaningful. Both distribution-matching and non-distribution-based metrics provide robust assessments when applied to a representative set of relevant features. For example, metrics like Jensen-Shannon divergence and Wasserstein distance are particularly effective for measuring alignment between synthetic and real data distributions. When applied to feature representations that capture relevant patterns, these metrics remain computationally efficient while offering meaningful insights into the fidelity of synthetic data in replicating real data characteristics.

Examples of features that can be extracted from images include radiomics (e.g., volume, sphericity, compactness), texture features (e.g., gray-level co-occurrence matrix, entropy), wavelet-based features, geometric descriptors (e.g., elongation, irregularity), and topological features. Additionally, deep learning-based features derived through unsupervised approaches can capture high-level complex patterns. For non-image data, relevant features include ICD-10 codes or SNOMED concepts in clinical notes, as well as clinical or laboratory data such as vital signs, lab results, etc. Feature correlation and summarization methods can further refine these extracted features by eliminating noise and redundancy, ensuring metrics focus on meaningful patterns rather than superficial alignment of distributions.

After extracting a good set of features, quantification can then be conducted by combining relevant features with a comprehensive set of metrics. This evaluation can occur at multiple levels—global (e.g., overall distribution) and local (e.g., specific regions of interest). For instance, metrics of Correctness and Coverage can be applied globally to assess overall alignment between real and synthetic data and overall diversity. Locally, Correctness and Coverage metrics can be applied to specific regions of interest, such as tumor areas, to evaluate the similarity or alignment between synthetic and real data in localized regions, as well as the diversity of features within those regions. This dual-level approach ensures that synthetic data captures not only overall patterns but also localized characteristics relevant to specific clinical regions.

In conclusion, while medical data is inherently high-dimensional, combining feature extraction, dimensionality reduction, and diverse set of quantitative metrics allows for effective evaluation of synthetic datasets. We are releasing a library (https://github.com/alzazmigliaa/SMD_ScoreCard) to support this framework by providing methods for extracting relevant features, feature summarization, and metrics of the 7Cs criteria.

Reviewer #2, comment #4

Correctness: I'm not sure I understand the relevance of BLEU score (LLMs), but perhaps that is because I do not understand the scope of data being covered as said in an earlier point

Our response #2.4

We acknowledge that the mixed examples provided may have caused some confusion. We apologize for the confusion. To address the reviewer's concern, we have revised the manuscript to enhance clarity.

Reviewer #2, comment #5

For all the principles: the framework is qualitative but all the principles are exemplified by quantitative metrics, what is a concrete recommendation for how to turn multiple possible metrics into for example "low coverage" assessment?

Our response #2.5

We thank the reviewer for raising this excellent point. We addressed this comment in the Discussion section (6th paragraph).

Specifically, we discussed how aggregating metrics into a single score is a well-established practice in various domains, and similar methodologies can be applied to synthetic medical data evaluation. For instance, credit scores combine multiple scores from factors—such as payment history, credit utilization, and account age—into a single score using methods like weighted averages or statistical models. Drawing from this approach, we propose aggregating multiple metrics within a specific dimension, such as Coverage, into an aggregated score to represent that dimension and determine whether the dataset demonstrates sufficient Coverage.

For example, to compute an aggregated score for a dimension like Coverage, several methods can be employed, including a normalized average, weighted average, geometric mean, or PCA. The normalized average scales each metric to a common range before averaging. Weighted averages assign greater importance to metrics based on their relevance to the application. The geometric mean provides a multiplicative approach that minimizes the impact of extreme values. PCA, in contrast, reduces dimensionality by focusing on the most informative combinations of metrics.

After calculating the aggregated score, converting into a concrete recommendation requires applying methods such as thresholding. Thresholds can be established based on clinical knowledge or reference patient dataset(s). For example, if patient datasets exhibit variability between 80–85, recommendations might be structured as follows: good Coverage: score ≥ 80 ; moderate Coverage: $70 \leq \text{score} \leq 80$; low Coverage: score < 70 . For situations where thresholds cannot be derived from patient data, expert consensus or standards can provide alternative.

While the methodology for computing aggregated scores can remain consistent across tasks, the thresholds are likely to vary depending on the application. This is analogous to the variability in acceptable sensitivity ranges for model performance across clinical tasks (e.g., cancer screening versus general health assessments). We hope this discussion clarifies how our framework can provide concrete recommendations for different criteria.

Reviewer #2, comment #6

Who should be doing these evaluations? Given the level of for example correctness, coverage and documentation of some existing public, non-synthetic datasets, it seems that researchers are not incentivized to do this.

Our response #2.6

Thanks a lot for making this comment, it helps us clarified this point in the manuscript (last paragraph in Discussion section).

As usage of synthetic data grows, the need for consistent, high-quality assessments of synthetic data quality will become increasingly important. A systematic framework for evaluating and reporting the quality

of synthetic datasets will enable anyone working with synthetic data to ensure that the data they rely on meets the necessary standards, ultimately improving the robustness of their models and the outcomes of their research. We anticipate that journals, funding agencies, and regulatory bodies will begin to require this type of dataset evaluation, further incentivizing adoption of this framework. Our goal in proposing this framework, along with an initial library and tool, was to provide a starting point that can be generally adopted by any stakeholders as needed.

To support the adoption of our framework, we are releasing an initial library that implements the proposed scorecard. Our goal is to provide a user-friendly and efficient solution to facilitate the adoption of the framework and promote its broader use across research and regulatory domains.

Thank you for your comments! We have addressed them in the manuscript (highlighted in blue) and discuss them above. This has improved the clarity of the manuscript.

Reply to the Reviewers

Re: Manuscript ID COMMS-ENG-24-0115-T

“Scorecards for Synthetic Medical Data Evaluation and Reporting”

Nature Communications Engineering

We thank the editor for handling our manuscript and both the editor and reviewers for their constructive comments, which helped us enhance the quality and clarity of our manuscript. We have addressed all the comments in the revised manuscript and used blue text to highlight the changes. Below, we provide a point-by-point response to each comment.

The revised manuscript contains minor editorial changes that were introduced as part of the internal clearance of the content by the FDA.

Reviewer 1: Overall Summary

This paper introduces the use of scorecards to assess synthetic medical data. It uses 7 'C's to produce this scorecard that cover the key concepts required in assessing a dataset.

This is an important topic and the scorecard seems like a useful approach to providing simple guidance on dataset quality. The 7 'C's appear to cover many of the key issues in synthetic data assessment.

Reviewer #1, comment #1

One issue I would like to see discussed is that of other types of data beyond images - which seems to be the sole focus of this paper. Medical data comes in many forms and these scorecards should naturally extend to medical notes and numerical clinical data. There is one sentence on LLMs in the discussion but I feel that more details in this section would make this a more useful paper in terms of covering the diversity of medical data. For example, a discussion on linking the specific issues with medical notes / clinical data to the scorecard approach.

Our response #1.1

We thank the reviewer for acknowledging the value of our work. We agree that the scorecard approach extends beyond medical images to other types of medical data, such as clinical notes and numerical clinical data. To address this, we have revised the manuscript to include examples illustrating how the scorecard can be applied to other forms of medical data. Additionally, we expanded the discussion to explain how each scorecard criterion is applicable to different data types.

In response to the reviewer's comment, we have made the following changes (highlighted in blue):

- Added an example of medical notes alongside medical images in Figure 1.
- Expanded the description of all criteria to make them general and applicable to other data types.
- Included three new paragraphs in the discussion (second, third, and fourth paragraphs) elaborating on how the same criteria can be applied to other forms of medical data.

We hope this revision clarifies that the scorecard approach is flexible to cover other types of medical data.

Reviewer #1, comment #2

Another issue is the universality of this approach. For example, regulations may depend on the use case, or the country that data is being used. As a result, the scorecard will change depending on these. This is not a major problem for the approach, but rather this should be explored and discussed within the discussion section.

Our response #1.2

We thank the reviewer for raising the important point of how regulations may vary depending on the use case or country. We agree that regulatory requirements can differ. To address this, we have revised Section 2 (Compliance) and Section 4 (fourth paragraph) to emphasize that the scorecard is designed to be adaptable. Just as different models may have unique performance requirements based on their specific applications, the scorecard criteria (such as Correctness, Coverage, and Compliance) can be applied across synthetic datasets but with tailored requirements based on the region or intended use case. Specifically, the Compliance criterion can be adjusted to meet local regulatory requirements. For example, Compliance in the U.S. may emphasize HIPAA standards while in Europe, the focus would shift to GDPR.

The Compliance criterion can be measured using the same metrics across regions and institutions while allowing flexibility in thresholds to meet specific requirements. For example, in privacy-preserving metrics like Differential Privacy (ϵ) and K-Anonymity, in contexts that require stricter standards, we might require lower ϵ values (e.g., $\epsilon = 0.5$) or higher k levels (e.g., $k \geq 10$) to minimize re-identification risks. In contrast, in other contexts with more flexible standards, we might permit higher ϵ values (e.g., $\epsilon = 2$) or lower k levels (e.g., $k = 5$) to balance utility and privacy. Similarly, standardization metrics, such as a DICOM Conformance Score, can universally evaluate adherence to medical imaging standards but emphasize different attributes depending on institutional priorities.

Reviewer 2, overall comment

The paper proposes scorecards, a systematic framework for generating synthetic medical data (SMD) which could be used for training and evaluating healthcare AI algorithms.

Given the increased use (and perhaps hype) about SMD it is crucial to evaluate its quality and I support the authors' effort towards creating this qualitative evaluation framework.

A couple of thoughts/suggestions that could be helpful to discuss:

Reviewer #2, comment #1

There is an important distinction between using SMD for training vs testing algorithms. In training, even noisy/not highly realistic data could still be helpful for the algorithm to be more robust. But using SMD for evaluation is ill-advised, as since the data is generated from the training distribution, in a sense the model would be overfitting to the training set.

Perhaps it would be good to give some specific examples of for example “garbage in garbage” and exemplify by specific research studies where this went wrong.

Our response #2.1

We appreciate the reviewer’s pointing out this important distinction between using SMD for training versus testing algorithms. We agree that the impact of synthetic data quality is more critical when used for testing due to overfitting. However, we believe it is still important to assess the quality of synthetic data used for training. We presented our reason below and we revised the manuscript (7th paragraph in the Discussion) to discuss the importance of ensuring the quality for training and testing.

Importance of assessing training data: While the quality of SMD is more critical when used for testing, it is also important to assess the quality of synthetic data used for training. Poor-quality synthetic data can mislead models by causing them to learn from irrelevant patterns or features, resulting in overfitting and poor generalization to real-world data. This issue was exemplified in the failure of IBM’s Watson for Oncology. This high-profile collapse was attributed to the reliance on synthetic data that failed to capture the complexities and variabilities of patient data. Similarly, a recent study (Shumailov et al.) revealed that LLMs tend to degrade when trained on recursively generated data, further emphasizing the need to scrutinize the quality of synthetic data used in training. Such proactive assessment of synthetic data prior to training is important for identifying potential limitations and addressing them early. Although synthetic data can introduce diversity, poor-quality data can negatively impact training by embedding unrealistic biases, confusing the model, or distracting it from learning relevant patterns. This highlights the importance of conducting pre-training quality checks.

- Shumailov, Ilya, et al. “AI models collapse when trained on recursively generated data.” *Nature* 631.8022 (2024): 755-759.
- Faheem, Hadiya, and Sanjib Dutta. “Artificial Intelligence Failure at IBM’s Watson for Oncology.” *IUP Journal of Knowledge Management* 21.3 (2023): 47-75.

Garbage in, Garbage out: In response, we have included specific examples in the revised manuscript to demonstrate how poor data quality negatively impacts machine learning models. These are highlighted in blue in the revised manuscript.

A notable real-world example of “garbage in, garbage out” is IBM’s Watson for Oncology, which performed poorly when deployed in clinical settings. One of the major reasons that leads to its failure, as reported in the paper below, was the use of synthetic datasets that lacked quality and failed to represent the variability of real-world data. Beyond this example, several research studies (e.g., Shumailov et al., Jordon et al., Ghassemi et al.) have shown how the quality of synthetic data can degrade utility and reduce downstream classification performance. This real world example along with these studies emphasize the principle of “garbage in, garbage out,” illustrating how poor-quality data can harm model performance.

This underscores the importance of proactive approaches to ensure the quality of generated synthetic data before its use in training or testing. Our proposed framework addresses this need by providing a systematic way to evaluate synthetic data quality, enabling early detection of potential issues.

- Shumailov, Iliia, et al. "AI models collapse when trained on recursively generated data." *Nature* 631.8022 (2024): 755-759.
- Faheem, Hadiya, and Sanjib Dutta. "Artificial Intelligence Failure at IBM'Watson for Oncology'." *IUP Journal of Knowledge Management* 21.3 (2023): 47-75.
- Cheng, Victoria, et al. "Can you fake it until you make it? impacts of differentially private synthetic data on downstream classification fairness." *Proceedings of the 2021 ACM Conference on Fairness, Accountability, and Transparency*. 2021.
- Jordon, James, et al. "Synthetic Data—what, why and how?." arXiv preprint arXiv:2205.03257 (2022).

Reviewer #2, comment #2

In fact while the scorecard intuitively makes sense to me, "healthcare AI algorithms" can include many things. Or is this limited to medical imaging, as Fig 1 might suggest? It would be helpful to see an example of a filled in model card for an existing dataset/algorithm, like in the Model Cards paper, or here <https://arxiv.org/abs/1803.09010>, or even specifically for health: <https://dl.acm.org/doi/fullHtml/10.1145/>

Our response #2.2

We thank the reviewer for raising these important points.

Scope beyond medical imaging: We have revised the manuscript to clarify how each criterion can be adapted to other data types. Specifically, we expanded Figure 1, elaborated on each criterion in Section 2 to illustrate these extensions, and added three paragraphs in the Discussion section.

While Figure 1 focuses on an example from medical imaging, we intend the scorecard to be adaptable and not limited to this domain. It can be applied across a variety of healthcare AI algorithms, including large language models (LLMs) for clinical notes, or numerical clinical data, and more. The criteria—such as Correctness, Coverage, and Compliance—are broad enough to evaluate AI systems across diverse healthcare contexts. This mirrors the flexibility of the model card framework, which is designed to be general and adaptable, with task-specific nuances.

Comparison to Model Card: As discussed in Section 3 of the revised manuscript, the Synthetic Medical Data (SMD) Card distinguishes itself from Healthsheets and Model Card by focusing explicitly on reporting the characteristics and quality (7Cs framework) of synthetic datasets. While the main concept of documenting and standardizing information about datasets or models is shared, our SMD Card is uniquely tailored to synthetic data. It emphasizes evaluating the quality of synthetic medical data across technical, clinical, and regulatory dimensions, providing a systematic and multi-modal approach.

Unlike Healthsheets or Model Cards, which typically focus on broader datasets or model characteristics, the SMD Card integrates quantitative metrics and numerical results to assess the dataset's performance in areas such as Correctness, Coverage, Completeness, and Constraint. These metrics provide actionable conclusions, allowing users to determine the suitability of synthetic datasets for specific applications. Further, similar to how the main sections and criteria for Model Cards and Healthsheets remain consistent across applications while details like sensitivity or specificity acceptance thresholds vary, the SMD Card maintains consistent core criteria but adapts its details to align with the requirements of synthetic data in different tasks.

SMD Card example: As per the reviewer's suggestion, we have made the following revisions: 1) a template of the SMD scorecard is now provided in Table 1 of the revised manuscript; 2) an example of a completed scorecard, using the Sinkove synthetic digital mammography dataset, has been included in our webpage (https://github.com/alzanzmigaa/SMD_scoreCard/tree/main/Sinkove_Example) to demonstrate its application; we did not provide the example in the manuscript as this is a perspective paper with limited word count; and 3) an example of our internal tool for generating the scorecard is shown on the project's webpage. Please note that this tool will be made publicly available.

Reviewer #2, comment #3

This is example is for Correctness, but I think I have similar concerns for all the principles. I agree that medical data is not small computer vision and that many metrics may not be appropriate. But the Correctness principle then talks about other metrics for matching distributions, it is unclear from the reading why these would be superior. Since the data is high-dimensional, I imagine it might be intractable to compare the distribution of images in their original resolution, and the data samples are not limited. So it is estimations of the distributions that will be compared – which means that it would be potentially possible for the distributions to be matching.

Our response #2.3

We thank the reviewer for their valuable feedback. In response, we clarified (Discussion section) that while Correctness (or Congruence) focuses on measuring alignment between synthetic and real data distributions to ensure a baseline level of similarity and alignment, other criteria, such as Coverage, extend beyond alignment to assess variability that reflects real-world diversity. Additionally, Constraint evaluates adherence to established clinical constraints specific to the medical task or modality. For example, Congruence ensures that synthetic data aligns with the statistical properties of real data, Coverage then evaluates the diversity and representativeness of the synthetic data using, and Constraint measures the adherence of synthetic data to known constraints, such as anatomical accuracy. For instance, Constraint could quantify whether tumors in synthetic images exhibit clinically accurate properties such as appropriate size, location, and shape, etc. Together, these criteria (along with the other four criteria) ensure a holistic assessment of synthetic data quality beyond distribution-matching between synthetic and patient data.

We also discussed the importance of extracting clinically relevant features instead of using the original resolution followed by using dimensionality reduction or feature summarization.

Detailed response: We agree that medical data differs from standard computer vision tasks, making direct comparisons of image distributions in their original resolution impractical. Metrics should instead focus on carefully extracted relevant features rather than raw image resolutions or generic features, ensuring evaluations are computationally feasible and clinically meaningful. Both distribution-matching and non-distribution-based metrics provide robust assessments when applied to a representative set of relevant features. For example, metrics like Jensen-Shannon divergence and Wasserstein distance are particularly effective for measuring alignment between synthetic and real data distributions. When applied to feature representations that capture relevant patterns, these metrics remain computationally efficient while offering meaningful insights into the fidelity of synthetic data in replicating real data characteristics.

Examples of features that can be extracted from images include radiomics (e.g., volume, sphericity, compactness), texture features (e.g., gray-level co-occurrence matrix, entropy), wavelet-based features, geometric descriptors (e.g., elongation, irregularity), and topological features. Additionally, deep learning-based features derived through unsupervised approaches can capture high-level complex patterns. For non-image data, relevant features include ICD-10 codes or SNOMED concepts in clinical notes, as well as clinical or laboratory data such as vital signs, lab results, etc. Feature correlation and summarization methods can further refine these extracted features by eliminating noise and redundancy, ensuring metrics focus on meaningful patterns rather than superficial alignment of distributions.

After extracting a good set of features, quantification can then be conducted by combining relevant features with a comprehensive set of metrics. This evaluation can occur at multiple levels—global (e.g., overall distribution) and local (e.g., specific regions of interest). For instance, metrics of Correctness and Coverage can be applied globally to assess overall alignment between real and synthetic data and overall diversity. Locally, Correctness and Coverage metrics can be applied to specific regions of interest, such as tumor areas, to evaluate the similarity or alignment between synthetic and real data in localized regions, as well as the diversity of features within those regions. This dual-level approach ensures that synthetic data captures not only overall patterns but also localized characteristics relevant to specific clinical regions.

In conclusion, while medical data is inherently high-dimensional, combining feature extraction, dimensionality reduction, and diverse set of quantitative metrics allows for effective evaluation of synthetic datasets. We are releasing a library (https://github.com/alzanzmigaa/SMD_ScoreCard) to support this framework by providing methods for extracting relevant features, feature summarization, and metrics of the 7Cs criteria.

Reviewer #2, comment #4

Correctness: I'm not sure I understand the relevance of BLEU score (LLMs), but perhaps that is because I do not understand the scope of data being covered as said in an earlier point

Our response #2.4

We acknowledge that the mixed examples provided may have caused some confusion. We apologize for the confusion. To address the reviewer's concern, we have revised the manuscript to enhance clarity.

Reviewer #2, comment #5

For all the principles: the framework is qualitative but all the principles are exemplified by quantitative metrics, what is a concrete recommendation for how to turn multiple possible metrics into for example "low coverage" assessment?

Our response #2.5

We thank the reviewer for raising this excellent point. We addressed this comment in the Discussion section (6th paragraph).

Specifically, we discussed how aggregating metrics into a single score is a well-established practice in various domains, and similar methodologies can be applied to synthetic medical data evaluation. For instance, credit scores combine multiple scores from factors—such as payment history, credit utilization, and account age—into a single score using methods like weighted averages or statistical models. Drawing from this approach, we propose aggregating multiple metrics within a specific dimension, such as Coverage, into an aggregated score to represent that dimension and determine whether the dataset demonstrates sufficient Coverage.

For example, to compute an aggregated score for a dimension like Coverage, several methods can be employed, including a normalized average, weighted average, geometric mean, or PCA. The normalized average scales each metric to a common range before averaging. Weighted averages assign greater importance to metrics based on their relevance to the application. The geometric mean provides a multiplicative approach that minimizes the impact of extreme values. PCA, in contrast, reduces dimensionality by focusing on the most informative combinations of metrics.

After calculating the aggregated score, converting into a concrete recommendation requires applying methods such as thresholding. Thresholds can be established based on clinical knowledge or reference patient dataset(s). For example, if patient datasets exhibit variability between 80–85, recommendations might be structured as follows: good Coverage: score ≥ 80 ; moderate Coverage: $70 \leq \text{score} \leq 80$; low Coverage: score < 70 . For situations where thresholds cannot be derived from patient data, expert consensus or standards can provide alternative.

While the methodology for computing aggregated scores can remain consistent across tasks, the thresholds are likely to vary depending on the application. This is analogous to the variability in acceptable sensitivity ranges for model performance across clinical tasks (e.g., cancer screening versus general health assessments). We hope this discussion clarifies how our framework can provide concrete recommendations for different criteria.

Reviewer #2, comment #6

Who should be doing these evaluations? Given the level of for example correctness, coverage and documentation of some existing public, non-synthetic datasets, it seems that researchers are not incentivized to do this.

Our response #2.6

Thanks a lot for making this comment, it helps us clarified this point in the manuscript (last paragraph in Discussion section).

As usage of synthetic data grows, the need for consistent, high-quality assessments of synthetic data quality will become increasingly important. A systematic framework for evaluating and reporting the quality

of synthetic datasets will enable anyone working with synthetic data to ensure that the data they rely on meets the necessary standards, ultimately improving the robustness of their models and the outcomes of their research. We anticipate that journals, funding agencies, and regulatory bodies will begin to require this type of dataset evaluation, further incentivizing adoption of this framework. Our goal in proposing this framework, along with an initial library and tool, was to provide a starting point that can be generally adopted by any stakeholders as needed.

To support the adoption of our framework, we are releasing an initial library that implements the proposed scorecard. Our goal is to provide a user-friendly and efficient solution to facilitate the adoption of the framework and promote its broader use across research and regulatory domains.

Thank you for your comments! We have addressed them in the manuscript (highlighted in blue) and discuss them above. This has improved the clarity of the manuscript.